# Anti-biofilm Fe_3_O_4_@C_18_-[1,3,4]thiadiazolo[3,2-*a*]pyrimidin-4-ium-2-thiolate Derivative Core-shell Nanocoatings

**DOI:** 10.3390/ma13204640

**Published:** 2020-10-17

**Authors:** Rodica Olar, Mihaela Badea, Cătălin Maxim, Alexandru Mihai Grumezescu, Coralia Bleotu, Luminiţa Măruţescu, Mariana Carmen Chifiriuc

**Affiliations:** 1Department of Inorganic Chemistry, Faculty of Chemistry, University of Bucharest, 90–92 Panduri Str., 050663 Bucharest, Romania; mihaela.badea@chimie.unibuc.ro (M.B.); catalin.maxim@chimie.unibuc.ro (C.M.); 2Department of Science and Engineering of Oxidic Materials and Nanomaterials, Faculty of Applied Chemistry and Materials Science, University Politehnica of Bucharest, 1–7 Polizu Street, 011061 Bucharest, Romania; grumezescu@yahoo.com; 3Stefan S Nicolau Institute of Virology, Romanian Academy, 285 Mihai Bravu Ave., 030304 Bucharest, Romania; coralia.bleotu@virology.ro; 4Department of Microbiology, Faculty of Biology, University of Bucharest, 1–3 Aleea Portocalelor St., 60101 Bucharest, Romania; luminita.marutescu@bio.unibuc.ro; 5Environment and Earth Sciences Department, Research Institute of the University of Bucharest—ICUB, Splaiul Independenţei 91–95, 050095 Bucharest, Romania; 6Academy of Romanian Scientists, 010071 Bucharest, Romania

**Keywords:** [1,3,4]thiadiazolo[3,2-*a*]pyrimidine, molecular structure, coated catheter, Fe_3_O_4_@C_18_, biofilm, cytotoxicity

## Abstract

The derivatives 5,7-dimethyl[1,3,4]thiadiazolo[3,2-*a*]pyrimidin-4-ium-2-thiolate (**1**) and 7-methyl-5-phenyl[1,3,4]thiadiazolo[3,2-*a*]pyrimidin-4-ium-2-thiolate (**2**) were fully characterized by single-crystal X-ray diffraction. Their supramolecular structure is built through both π–π stacking and C=S–π interactions for both compounds. The embedment of the tested compounds into Fe_3_O_4_@C_18_ core-shell nanocoatings increased the protection degree against *Candida albicans* biofilms on the catheter surface, suggesting that these bioactive nanocoatings could be further developed as non-cytotoxic strategies for fighting biofilm-associated fungal infections.

## 1. Introduction

Despite significant research progress in developing novel anti-infectious therapeutic and preventive strategies, infectious diseases treatment still remains a big challenge, primarily because of the emergence of microbial resistance to nearly all antibiotics used clinically [1,2,3]. The most threatening are those produced by the multiple drug-resistant (MDR) bugs that emerged in both hospitals and the community. The Gram-negative MDR bugs are intrinsically or clinically resistant to all currently available antibiotics (mediated by efflux pumps overexpression or by the presence of protective outer membrane), the resistance genes being often located on mobile genetic elements, facilitating their accumulation and dissemination [4].

Many efforts are directed towards searching either novel compounds with potent bioactivities or new ways of formulation or administration for achieving efficient tools to fight especially multi-resistant strains or biofilm formation on medical implants [5,6]. Once adhered and protected by a self-secreted extracellular polymeric matrix, microbial cells switch to a modified phenotype, with a different growth rate and transcriptome, becoming up to 1000-fold more tolerant to antibiotics. Usually, the active antibiotic concentrations are determined by standardized antimicrobial susceptibility tests performed on planktonic cells, which are not active against biofilms [7].

One of the most promising leads for new antimicrobial strategies is to design antimicrobials with dual/multiple mechanisms of action, hybrid antibiotics, which are defined as synthetic constructs with >2 pharmacophores with antimicrobial activity or of hybrid materials combining different antimicrobials in a single system [8,9,10]. In this way, the required active dose and the risk of resistance selection are minimized, increasing efficiency and reducing the side effects. 

Nanotechnology proposed various nanomaterials to fight resistant bacterial and biofilms, either based on their intrinsic antimicrobial (microbiostatic/microbicidal/antipathogenic) properties or on their potential to be used as delivery systems for antimicrobial drugs [5,9,11,12]. In this regard, the hybrid nanosystems based on magnetite nanoparticles (MGNPs) could represent an alternative, considering their accessible and versatile preparation, reduced toxicity as well as antimicrobial properties [13,14,15,16]. MGNPs were authorized as in vivo drug delivery platform, and as a result, several loaded systems with a good antimicrobial activity were recently reported [17,18,19,20,21,22,23].

On the other hand, a variety of biomolecules have been used in order to facilitate the interaction of MGNPs with other antimicrobial species [20], including fatty acids such as stearic and myristic ones [17,24].

Nitrogen-based heterocycles are most studied as biologically active species, and among these, the 1,3,4-thiadiazole moiety exhibit manifold importance as a potent chemotherapeutic agent, this being the active pharmacophore in several drugs. Thus acetazolamide and methazolamide are used for the treatment of glaucoma, megazol in *Tripanosoma brucei* and *T. cruzi* parasitoses, while sulfamethoxazole, cefazolin, and cefazedone are large spectrum antibiotics [25,26,27,28].

Moreover, thiadiazole is a versatile substrate for designing a large diversity of fused heterocycles, such as those with triazole [29,30,31,32,33,34], imidazole [35,36,37,38,39], and pyrimidine [40,41,42,43,44,45,46,47,48,49,50,51,52], most of them with confirmed biological activity.

Among these, the thiadiazolopyrimidine fused systems are particularly interesting as mesoionic purine bases analogs. Otherwise, the [1,3,4]thiadiazolo [3,2-*a*]pyrimidine core induces a broad spectrum of pharmacological activities, including antimicrobial [40,41,42,43,44,45,46] and antitumoral ones [47,48,49,50,51,52].

Using a nanobiotechnological approach, we have combined the tools and materials of synthetic chemistry, nanoscience, and biology to address the antimicrobial resistance threat. As a result, this paper focuses on the structural characterization and evaluation of the antibiofilm activity of two bioactive nanosystems based on [1,3,4]thiadiazolo [3,2-*a*]pyrimidinium derivatives and Fe_3_O_4_@C_18_ core@shell nanoparticles against *Staphylococcus aureus*, *Pseudomonas aeruginosa* and *Candida albicans* strains in order to develop a new strategy for preventing both formation and eradication of microbial biofilms formed on medical devices. This research is based on previous data concerning the antimicrobial activity of MGNPs [13,14,15,16] and synergy observed with antimicrobials [17,18,19,20,53,54], a strategy that could be used in order to develop valuable nanocarriers for a controlled or a targeted release of an active agent. The interaction between the MGNPs and the [1,3,4]thiadiazolo[3,2-*a*]-pyrimidinium derivatives was facilitated by octadecanoic acid (C_18_). Furthermore, the crystallographic data concerning 5,7-dimethyl[1,3,4]thiadiazolo[3,2-*a*]pyrimidin-4-ium-2-thiolate (**1**) and 7-methyl-5-phenyl[1,3,4]thiadiazolo[3,2-*a*]pyrimidin-4-ium-2-thiolate (**2**) are reported together with their cytotoxicity against HeLa cells.

## 2. Experimental 

### 2.1. General Information

The reagents of high purity grade were purchased from Merck (Darmstadt, Germany, pentane-2,4-dione, 1-phenyl-1,3-butanedione), Fluka (Fluka Chemical-Sigma Aldrich, Taufkirchen, Germany, 5-amino-1,3,4-thiadiazole-2-thiol), and Sigma Aldrich (Munich, Germany, FeCl_3_ and FeSO_4_⋅7H_2_O), and used as received without further purification. 

The C, N, and H content was obtained by a Perkin Elmer PE 2400 analyzer (Perkin Elmer, Waltham, MA, USA). FT-IR spectra in KBr pellets were recorded by a Bruker Tensor 37 spectrometer (Bruker, Billerica, MA, USA) (400–4000 cm^−1^ range). ^1^H-NMR and ^13^C-NMR spectra were recorded at 25 °C on a Bruker Avance spectrometer (Bruker, Karlsruhe, Germany) (working frequency 200 MHz). The chemical shifts (parts per million) were related to the tetramethylsilane (TMS) as an internal standard.

### 2.2. Synthesis and Spectral Data for [1,3,4]thiadiazolo[3,2-*a*]pyrimidine Derivatives (Adsorption-Shell)

In order to synthesize the compounds, a modified procedure was used [55]. To a solution containing pentane-2,4-dione/1-phenyl-1,3-butanedione (0.02 mol) in ethanol (25 mL), 5-amino-1,3,4-thiadiazole-2-thiol (0.02 mol) in ethanol (25 mL) was added, followed by a few drops of CH_3_COOH. The reaction mixture was refluxed 12 h when upon a yellow solution was formed. The solution volume was reduced at the half, and the yellow solid product formed was filtered off, washed several times with cold EtOH and ethyl ether. Single crystals suitable for X-ray diffraction were isolated by the slow evaporation of an ethanolic solution.

*5,7-Dimethyl[1,3,4]thiadiazolo[3,2-a]pyrimidin-4-ium-2-thiolate* (**1**) Analysis found: C, 42.64; H, 3.49; N, 21.42; S, 32.59, C_7_H_7_N_3_S_2_ requires: C, 42.62; H, 3.58; N, 21.30; S, 32.51; Yield 93%; IR (KBr pellet, cm^−1^): υ(CH pym), 3011w; υ_as_(CH_3_), 2962w; υ_s_(CH_3_), 2914w; υ(C=N)+υ(C=C), 1602vs, 1522s; δ_as_(CH_3_), 1418m; δ_s_(CH_3_), 1367s; TII, 1340vs, TIII, 1053vs; υ(N–N), 957m; TIV 715w; υ_as_(C–S–C), 645w; υ_s_(C–S–C), 535w [56]; ^1^H-NMR (DMSO-d_6_) δ (ppm): 2.64 (s, CH_3_), 2.83 (s, CH_3_), 7.19 (s, CH pyrimidine); ^13^C-NMR (DMSO) δ (ppm): 24.5 (CH_3_), 77.2 (C2 thiadiazole), 117.8 (C6 pyrimidine), 149.9 (C9 pyrimidine), 163.8 (C7 pyrimidine), 165.6 (C5 pyrimidine). 

*7-Methyl-5-phenyl[1,3,4]thiadiazolo[3,2-a]pyrimidin-4-ium-2-thiolate* (**2**) Analysis found: C, 55.64; H, 3.49; N, 16.32; S, 24.79, C_12_H_9_N_3_S_2_ requires: C, 55.57; H, 3.50; N, 16.20; S, 24.73; Yield 88%; IR (KBr pellet, cm^−1^): υ(CH aromatic), 3069w; υ(CH pyrimidine), 3011w; υ_as_(CH_3_), 2962w; υ_s_(CH_3_), 2914w; υ(C=N)+υ(C=C), 1600vs, 1570s, 1512s; δ_as_(CH_3_), 1418m; δ_s_(CH_3_), 1367s; TII, 1334vs, υ(C–CH), 1266w TIII, 1023vs; υ(N–N), 957m; ρ(CH), 844w, 770w, 660w, TIV 700m; υ_as_(C–S–C), 637w; υ_s_(C–S–C), 546w [56]; ^1^H-NMR (DMSO-d_6_) δ (ppm): 2.72 (s, CH_3_), 7.33 (s, CH pyrimidine), 7.54 (multiplet, Ar-H), 7.93 (d, Ar-H); ^13^C-NMR (DMSO) δ (ppm): 24.7 (CH_3_), 77.4 (C2 thiadiazole), 117.3 (C6 pyrimidine), 129.1, 129.4, 130.1, 132.8 (C10–C15; Ar), 149.1 (C9 pyrimidine), 165.2 (C7 pyrimidine), 165.7 (C5 pyrimidine).

### 2.3. X-ray Crystallography

Crystallographic data were collected with an IPDS II diffractometer (STOE, Darmstadt, Germany) having a Mo-Kα (λ = 0.71073 Å) X-ray tube with a graphite monochromator. A crystal of suitable size was selected from the mother liquor and immersed in paratone oil, then mounted on the tip of glass fiber and cemented using epoxy resin. Data collections: Stoe X-AREA. Cell refinement: Stoe X-AREA [57]. The structures were solved by direct methods and refined with anisotropic displacement parameters based on F^2^, using SHELXS 97 [58] and SHELXL 97 [59] programs. Packing and H-bonding diagrams are generated by the DIAMOND program. Non-hydrogen atoms were anisotropically refined until convergence was reached. CCDC-1986171 (**1**) and -1986172 (**2**) contain the supplementary crystallographic data that can be obtained free of charge from the Cambridge Crystallographic Data Centre via www.ccdc.cam.ac.uk/data_request/cif.

### 2.4. Synthesis and Characterization of Core@Shell Nanostructure

The core@shell—Fe_3_O_4_@C_18_ nanostructure was obtained as previously reported [17]. Briefly, the octadecanoic acid (C_18_) was dispersed in 200 mL volume of deionized water, corresponding to a 0.50% (v/w) solution, under vigorous stirring at 60 °C. 5 mL of 25% ammonia solution was added to C_18_ solution and then 200 mL of deionized water containing 1 g of FeCl_3_ and 1.6 g of FeSO_4_⋅7H_2_O (2:1 molar ratio) were dropwise added, under continuous stirring up to the formation of a black precipitate. The product was washed several times with methanol and deionized water, and separated with a strong NdFeB permanent magnet. Further, the samples were characterized by TEM (FEI Company, Hillsboro, OR, USA), XRD (Shimadzu, Kyoto, Japan) and FTIR (Thermo Nicolet, Madison, WI, USA), as previously reported by the authors [60,61,62].

### 2.5. The Core@Shell@Adsorption-Shell Nanostructure Deposition on Catheter Samples

The adsorption-shell nanostructure was obtained by grounding 1 mL solution of acetone containing 5 mg of (**1**) or (**2**) with 95 mg Fe_3_O_4_@C_18_ up to the complete solvent evaporation (Figure 1). This procedure was three times repeated for a uniform distribution of the samples on the surface of the spherical nanostructure. The catheters were then coated with the suspension of core@shell@adsorption-shell in acetone (0.33% w/v) and sterilized for 15 min by irradiation with ultraviolet light. This procedure was already reported elsewhere [63].

### 2.6. Screening of the Antibiofilm Activity

Fresh 24 h cultures were prepared on tryptone soya agar (TSA, MS 2000 Trading Impex SRL, Bucharest, Romania) from the microbial strains (*Staphylococcus aureus* ATCC 6538, *Pseudomonas aeruginosa* ATCC 27853, and *Candida albicans* ATCC 90029) (ATCC, Manassas, VA, USA) preserved on glycerol.

Biofilm development in 6-multi well plates (Nunc, Thermo Scientific, Antisel, Bucharest, Romania) was assessed in dynamics. Sterile coated and bare catheter (reference) samples were added in a 6-well plate in 2 mL of sterile saline and inoculated with ~10^5^ colony forming units (CFU)/mL of microbial suspensions. The samples were incubated at 37 °C for different periods of time periods (24, 48, and 72 h, respectively) to assess the temporal dynamics of cultures grown in the presence of coated and uncoated catheter samples. After each incubation time, the samples were taken off from the culture medium, gently washed to remove the non-adherent bacteria, sonicated to detach the adherent bacteria from the catheter surface, and the absorbance of the obtained suspension was measured at 600 nm with an Apollo LB 911ELISA reader (Berthold Technologies GmbH & Co. KG, Bad Wildbad, Germany).

### 2.7. Cytotoxicity Assay

HeLa cells (ATCC CCL-2, Manassas, VA, USA) were used to evaluate the toxicity of compounds. For this, 7.5 × 10^5^ cells/well were seeded in 24 wells plate, in 5% CO_2__,_ and humidified atmosphere at 37 °C for 24 h. The treatment using a concentration of compounds between 500 µg/mL and 125 µg/mL was evaluated after 24 h using CellTiter aqueous one solution kit (Promega, Madison, WI, USA), or using MitoTracker Red CMXRos (Thermo Fisher Scientific, Waltham, MA, USA), or was subjected to cell cycle analysis. The cells stained with 50 nM MitoTracker Red CMXRos, a fluorescent dye that stains mitochondria in live cells, were counterstained with 10 µg/mL Hoechst 33342. The stained cells were fixed in 4% paraformaldehyde and photographed using an Observer D1 microscope (Zeiss, Jena, Germany) using 546 nm (red) and 305 nm (blue) filter. For cell cycle analysis, treated and untreated cells were fixed in cold ethanol (70%), stained with 100 µg/mL propidium iodide and analyzed at flow cytometer (Beckman Coulter, Nyon, Switzerland) and analyzed by FlowJo™ Software for Windows, version 7.2.5. (Becton, Dickinson and Company; 2019).

## 3. Results and Discussion

### 3.1. Synthesis and Characterization of [1,3,4]thiadiazolo[3,2-a]pyrimidine Derivatives

The reaction in 1:1 molar ratio of acetylacetone/benzoylacetone and 5-amino-1,3,4-thiadiazole-2-thiol produced the species 5,7-dimethyl[1,3,4]thiadiazolo[3,2-*a*]pyrimidin-4-ium-2-thiolate (**1**) and 7-methyl-5-phenyl[1,3,4]thiadiazolo[3,2-*a*]pyrimidin-4-ium-2-thiolate (**2**) as depicted in Scheme 1.

The IR spectra of the compounds (see the Experimental section) display bands at 1600 and 1510 cm^−1^ characteristic for thiadiazolopyrimidine and pyrimidine ring vibrations. The bands around 3100, 1550, and 1440 cm^−1^ correspond to thiadiazole moiety vibrations.

In the ^1^H-NMR spectrum of 5,7-dimethyl[1,3,4]thiadiazolo[3,2-*a*]pyrimidin-4-ium-2-thiolate, the signals at 2.64, and 2.83 ppm are assigned to methyl protons while that at 7.19 arises from the CH group of the pyrimidine ring. For 7-methyl-5-phenyl[1,3,4]thiadiazolo[3,2-*a*]pyrimidin-4-ium-2-thiolate, these signals are slightly shifted while the additional ones at 7.54 and 7.93 ppm could be assigned to phenyl ring protons. The ^13^C-NMR spectra display signals characteristic for pyrimidine moiety around 118, 150, and 166 ppm while the methyl, thiadiazole, and phenyl groups could be identified through signals around 25, 77, and 128–132 ppm, respectively.

### 3.2. Structural Description of [1,3,4]thiadiazolo[3,2-a]pyrimidine Derivatives

A summary of the crystallographic data and details of data collection of (**1**) and (**2**) s given in Table 1, while selected bond lengths and angles are given in Table 2. Compound (**1**) crystallizes in the monoclinic system, space group *P2_1_/n*, with one independent molecule in the asymmetric unit (Figure 2).

Several peculiar features were observed when analyzing the bond lengths between the atoms of the pyrimidine ring as well as the thiadiazole C–S and N–N bonds. Thus, the C2–N2 bond length, equal to 1.366(2) Å, is an intermediary value between the length of a C–N bond in an aliphatic chain (usually between 1.472(5)–1.479(5) Å) and in a heterocycle (1.426(12) Å) [65]. The size of the C6–N1 bond of 1.315(2) Å, is smaller than the previous one and corresponds to a double bond length, similar to a distance of azomethine or oxadiazole. The C7–S1 (1.767(2) Å) and C6–S1 (1.7216(18) Å) bonds have representative lengths for simple covalent bonds. As well, an observation was made that the C7-S2 (1.7216(18) Å) bond lengths are slightly higher than a C=S bond length, which confirms the zwitterion model of the compound.

Interestingly, at the supramolecular level, the molecules form supramolecular dimers built by aromatic π–π stacking interactions between pyrimidine rings of two adjacent molecules (distances between centroids 3.557 Å) (Appendix A).

Further, these dimers are connected by C=S–π intermolecular interactions between the C(7)=S(2) groups and the pyrimidine rings (Figure 3) with distances of 3.429 Å, resulting in a 2D brick wall supramolecular network. This type of interaction is more common in protein structure chemistry [66,67,68,69,70,71]. The single-crystal X-ray diffraction molecular structure of compound (**2**) shows the same zwitterion structure with a similar pyrimidine and thiadiazole rings (Figure 4).

Compound (**2**) crystallizes in the monoclinic system, space group *P2_1_/c*, with one independent molecule in the asymmetric unit. The thiadiazole-pyrimidine (S1/N3–N2/C2–N1/C6) ring is essentially planar, with a maximum deviation of 0.089 Å for the methyl group. The bond lengths for the two rings are very similar with the first compound (C1–S2=1.684(5) Å, N2–N3=1.378(6) Å, C2–S1=1.761(5) Å, C2–N1=1.314(7) Å, C2–N2=1.374(6) Å). The dihedral angle between the thiadiazole-pyrimidine ring and the benzene ring is 34.691°.

The network stabilization for compound (**2**) comes from both weak π–π stacking interactions between the thiadiazole (S1/C1–N3/N2–C2) units (a centroid-to-centroid distance of 3.873 Å) and C=S–π intermolecular interactions between the C(1)=S(2) groups and the pyrimidine rings, with distances of 3.592 Å (Figure 5). These columns running along the *a* axis are packed in the crystal lattice (Appendix A) by non-covalent interactions which reinforce the crystal structure cohesion.

### 3.3. Antimicrobial Activity

The comparative analysis of Figure 6 and Figure 7 reveals that microbial biofilms were generally less developed in the presence of the catheter samples coated with the hybrid nanomaterials, as compared to the uncoated catheter or to the catheter coated with the free compounds, as indicated by the lower absorbance values of the microbial suspensions recovered from the catheter samples at different time points.

The bacterial and fungal biofilms grown in the presence of the uncoated catheter samples followed a different dynamic, recording a plateau in case of the three tested time points in the case of *S. aureus,* an exponential growth from 24 to 72 h for *P. aeruginosa*, and a decrease of biofilm mass installed gradually after 24 h for *C. albicans.*

Regarding the dynamics of microbial biofilms developed on catheter samples functionalized either with the two compounds or with the hybrid nanostructures respectively, although the growth trend line is relatively similar with that of control biofilm, however, a stronger inhibition of biofilm growth is evident in the case of the hybrid nanostructure.

The most significant biofilm inhibition is observed in case of *P. aeruginosa* biofilm grown on the catheter coated with the hybrid nanomaterials containing both compounds (**1**) (at all three time points) and (**2**) (mostly at 72 h). These results are significant, taking into account the high capacity of this opportunistic and nosocomial, multi-drug resistant pathogen to persist in the hospital environment and to produce biofilm-associated infections.

### 3.4. Cytotoxicity Assay

In order to estimate the cytotoxicity of the compounds, HeLa cells were exposed to the binary dilution of the species for 24 h and evaluated using CellTiter assay. Compound (**1**) demonstrated only weak toxicity at 500 µg/mL, while compound (**2**) was slightly more toxic. In the presence of (**2**), the cells died, and only a few adherent cells were stained with MitoTracker red. At 500 µg/mL of (**2**), the morphology of the cells was changed and the Hoechst dye staining was more intense, proving that those cells were dead (Figure 8).

The tested species did not exhibit any remarkable effects on the cellular cycle. Only the G2/M phase is slightly diminished at concentrations above 250 µg/mL (Table 3). The apoptosis peak is observed to the left of the G0/G1 phase and is proportional to the compound concentration (Figure 9). The differences in the cytotoxicity of these two compounds can be at least partially explained by the fact that the phenyl radical on the pyrimidine ring confers enhanced lipophilicity to the compound (**2**), and this can increase its ability to enter into the tumor cells.

It is worth mention that the catheter samples functionalized with the tested compounds did not have any cytotoxic effect on HeLa cells.

## 4. Conclusions

The species 5,7-dimethyl[1,3,4]thiadiazolo[3,2-*a*]pyrimidin-4-ium-2-thiolate (**1**) and 7-methyl-5-phenyl[1,3,4]thiadiazolo[3,2-*a*]pyrimidin-4-ium-2-thiolate (2) have been fully characterized by single-crystal X-ray diffraction that evidenced a zwitterion structure with π–π stacking and C=S–π interactions between species in solid network. The embedment of the tested compounds into Fe_3_O_4_@C_18_ core-shell nanocoating induced an increased anti-biofilm activity, reducing the colonization of the catheter surface with *C. albicans*, suggesting that these bioactive systems could be considered a promising biocompatible strategy foe efficiently fighting against biofilm-associated fungal infections.

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
