# Peer review of "Anti-biofilm Fe_3_O_4_@C_18_-[1,3,4]thiadiazolo[3,2-*a*]pyrimidin-4-ium-2-thiolate Derivative Core-shell Nanocoatings"

_materials, 2020, doi:10.3390/ma13204640_

Round 1
Reviewer 1 Report
The manuscript of Olar et al. reports the molecular and supramolecular structure of 5,7-dimethyl[1,3,4]thiadiazolo[3,2-a]pyrimidin-4-ium-2-thiolate and 7-methyl-5-phenyl[1,3,4]thiadiazolo[3,2-a]pyrimidin-4-ium-2-thiolate, with a particular focus on their properties as antimicrobial agents. The work reported in the manuscript looks interesting and valuable; the authors report a significant amount of new results, and the experimental data seem carefully acquired.
Some minor points are listed below:
- The authors should include the chemical structures of compounds 1 and 2 drawn in ChemDraw (or similar drawning software);
- The IUPAC name of stearic acid (octadecanoic acid) should be used at the first mention;
- The authors should provide atom numbering scheme for all atoms (not all atom numbers included in Table 1 are given in Figures 2 and 4);
- A figure with graphical representation/visualization of the Core@shell and Core@Shell@Adsorption-Shell nanostructures/hybrid nanomaterials would bring clarity and surely enrich the article.
Author Response
The answer for Reviewer 1:
- The authors should include the chemical structures of compounds 1 and 2 drawn in ChemDraw (or similar drawning software);
- The chemical structura in ChemDraw was added.
- The IUPAC name of stearic acid (octadecanoic acid) should be used at the first mention;
- The stearic acid was replaced by octadecanoic acid
- The authors should provide atom numbering scheme for all atoms (not all atom numbers included in Table 1 are given in Figures 2 and 4);
- The atom numbering scheme for all atoms was provided in Figures 1 and 3.
- A figure with graphical representation/visualization of the Core@shell and Core@Shell@Adsorption-Shell nanostructures/hybrid nanomaterials would bring clarity and surely enrich the article.
- A scheme was added.

Reviewer 2 Report
The manuscript is interesting and well - written, however, "new nanobiotechnological vision" in the title suggests more biological results.
Author Response
The answer for Reviewer 2:
The manuscript is interesting and well - written, however, "new nanobiotechnological vision" in the title suggests more biological results.
Title has been modified.

Reviewer 3 Report
In the paper entitled “Reviving old [1,3,4]thiadiazolo[3,2-a]pyrimidin-4-ium-2-thiolate derivatives in a new nanobiotechnological vision” the authors, Rodica Olar, Mihaela Badea, Cătălin Maxim, Alexandru Mihai Grumezescu, Coralia Bleotu, Luminiţa Măruţescu and Mariana Carmen Chifiriuc, developed a new nanoformulation for the administration of [1,3,4]thiadiazolo-based antimicrobial drugs.
The relevance of the topic is well discussed in the introduction section and supported by appropriate and recent citations. The scientific results have been discussed not in a balanced manner. For example:
- there is a wide description of the crystallographic data of the two drugs 1 and 2, but there is a poor description of the morphological properties of the nanoparticles (i.e. TEM or SEM analysis demonstrating the core-shell structure of the NPs and the adsorption of the drug)
- there is an appropriate description of the cell tests assessing the cytotoxicity and the antimicrobial activity of the drug@NPs, supported also by graphs, figures and a table; on the other hand, the drug synthesis and the preparation of the drug@NPs is so roughly discussed.
For these reasons I recommend this manuscript for the publication in Materials after the following major revisions:
#1 The abstract should be re-written taking into account the real focus of the paper (I do not think that the crystallographic data should be inserted in the abstract).
#2 Insert a Figure describing the structures of the drugs 1 and 2 and the synthetic procedures adopted to prepare them.
#3 Insert a Figure describing, also with a sketch, the structure of the drug@NPs tested in the paper.
#4 Improve the 3.3 paragraph by expanding the discussion on the structural characterization of the drug@NPs; the insertion of Figures is welcome.
#5 The 3.2 paragraph is too long and with too many figures; try to sum up it.
#6 Please insert a statistical analysis of data reported in Figures 7 and 8. Insert in the Figure 8 the time details as reported in Figure 7.
Author Response
The answer for Reviewer 3:
The relevance of the topic is well discussed in the introduction section and supported by appropriate and recent citations. The scientific results have been discussed not in a balanced manner. For example:
- there is a wide description of the crystallographic data of the two drugs 1 and 2, but there is a poor description of the morphological properties of the nanoparticles (i.e. TEM or SEM analysis demonstrating the core-shell structure of the NPs and the adsorption of the drug)
- there is an appropriate description of the cell tests assessing the cytotoxicity and the antimicrobial activity of the drug@NPs, supported also by graphs, figures and a table; on the other hand, the drug synthesis and the preparation of the drug@NPs is so roughly discussed.
#1 The abstract should be re-written taking into account the real focus of the paper (I do not think that the crystallographic data should be inserted in the abstract).
The abstract has been rewritten. The crystallographic data were removed from abstract.
#2 Insert a Figure describing the structures of the drugs 1 and 2 and the synthetic procedures adopted to prepare them.
A figure with the structures of the drugs 1 and 2 and the synthetic procedures used to prepare them was added (Scheme 1).
#3 Insert a Figure describing, also with a sketch, the structure of the drug@NPs tested in the paper.
A scheme describing the procedure was added.
#4 Improve the 3.3 paragraph by expanding the discussion on the structural characterization of the drug@NPs; the insertion of Figures is welcome.
Paragraph 3.3 was removed. All details are available in other papers cited in the document.
#5 The 3.2 paragraph is too long and with too many figures; try to sum up it.
The paragraph 3.2. was reduced and figures were moved in Supplementary materials. As result the other figures were renumbered. We preserved in the manuscript only the crystal packing for the two compounds, figures 2 and 4 due to the importance and rarity of such non-covalent interactions (C=S∙∙∙∙π) in supramolecular chemistry.
#6 Please insert a statistical analysis of data reported in Figures 7 and 8. Insert in the Figure 8 the time details as reported in Figure 7.
Figures 7 and 8 became 6 and 7 and have been modified as required.

Round 2
Reviewer 3 Report
I appreciate the work made by the authors in re-elaborating the paper. However, I have to point out a mismatch between the data reported in Figure 5 and 6 and the corresponding discussion text in section 3.3. This section starts with the statement “The comparative analysis of Figures 5 and 6 reveals that the hybrid nanomaterials exhibited a much more improved antibiofilm activity as compared to the free compound, as indicated the lower absorbance values of the microbial suspensions recovered from the catheter samples.”; but if you analyze the data reported in Figure 5 and 6 considering also the statistical analysis, you can observe that the only significant data are those from P. aeruginosa incubated with (1)+N at each time, and P. aeruginosa incubated with (2)+N at 72 h.
So, the antimicrobial activity of the hybrid nanomaterials should be scaled down and the text revised.
On the basis of these considerations, I will reconsider the submission of the paper after these revisions.
Author Response
We thank the reviewer for the careful revision.
We have reformulated the discussion part related to fig. 5 and 6 accordingly.
The new text is:
The comparative analysis of Figures 5 and 6 reveals that microbial biofilms had generally less developed in the presence of the catheter samples coated with the hybrid nanomaterials, as compared to the uncoated catheter or to the catheter coated with the free compounds, as indicated by the lower absorbance values of the microbial suspensions recovered from the catheter samples at different time points.
The bacterial and fungal biofilms grown in the presence of the uncoated catheter samples followed a different dynamic, recording a plateau in case of the three tested time points in the case of S. aureus, an exponential growth from 24 to 72 h for P. aeruginosa, and a decrease of biofilm mass installed gradually after 24 h for C. albicans.
Regarding the dynamics of microbial biofilms developed on catheter samples functionalized either with the two compounds or with the hybrid nanostructures respectively, although the growth trend line is relatively similar with that of control biofilm, however, a stronger inhibition of biofilm growth is evident in the case of the hybrid nanostructure.
The most significant biofilm inhibition is observed in case of P. aeruginosa biofilm grown on the catheter coated with the hybrid nanomaterials containing both compounds (1) (at all three time points) and (2) (mostly at 72h). These results are significant, taking into account the high capacity of this opportunistic and nosocomial, multi-drug resistant pathogen to persist in the hospital environment and to produce biofilm-associated infections.

Round 3
Reviewer 3 Report
In this form I accept the paper for the publication on Materials